# Dopamine D_2_ and Serotonin 5-HT_1A_ Dimeric Receptor-Binding Monomeric Antibody scFv as a Potential Ligand for Carrying Drugs Targeting Selected Areas of the Brain

**DOI:** 10.3390/biom12060749

**Published:** 2022-05-26

**Authors:** Agata Kowalik, Mateusz Majerek, Krzysztof Mrowiec, Joanna Solich, Agata Faron-Górecka, Olga Woźnicka, Marta Dziedzicka-Wasylewska, Sylwia Łukasiewicz

**Affiliations:** 1Department of Physical Biochemistry, Faculty of Biochemistry, Biophysics and Biotechnology, Jagiellonian University, 30-387 Krakow, Poland; agata.kowalik@doctoral.uj.edu.pl (A.K.); mateusz.majerek@student.uj.edu.pl (M.M.); krzysztof.mrowiec@student.uj.edu.pl (K.M.); marta.dziedzicka-wasylewska@uj.edu.pl (M.D.-W.); 2Department of Pharmacology, Maj Institute of Pharmacology, Polish Academy of Science, 31-343 Krakow, Poland; solich@if-pan.krakow.pl (J.S.); gorecka@if-pan.krakow.pl (A.F.-G.); 3Department of Cell Biology and Imaging, Institute of Zoology and Biomedical Research, Jagiellonian University, 30-387 Krakow, Poland; olga.woznicka@uj.edu.pl

**Keywords:** scFv antibody, D_2_–5-HT_1A_ heteromer, nanoparticles, nanomedicine, targeted drug delivery, nanogold particles, cytotoxicity

## Abstract

Targeted therapy uses multiple ways of ensuring that the drug will be delivered to the desired site. One of these ways is an encapsulation of the drug and functionalization of the surface. Among the many molecules that can perform such a task, the present work focused on the antibodies of single-chain variable fragments (scFvs format). We studied scFv, which specifically recognizes the dopamine D_2_ and serotonin 5-HT_1A_ receptor heteromers. The scFv_D2–5-HT1A_ protein was analyzed biochemically and biologically, and the obtained results indicated that the antibody is properly folded and non-toxic and can be described as low-immunogenic. It is not only able to bind to the D_2_–5-HT_1A_ receptor heteromer, but it also influences the cAMP signaling pathway and—when surfaced on nanogold particles—it can cross the blood–brain barrier in in vitro models. When administered to mice, it decreased locomotor activity, matching the effect induced by clozapine. Thus, we are strongly convinced that scFvD_2_–5-HT_1A_, which was a subject of the present investigation, is a promising targeting ligand with the potential for the functionalization of nanocarriers targeting selected areas of the brain.

## 1. Introduction

One of the fundamental goals of targeted therapies is to use a system that will allow the drug to be directed to its potential site of action. Drug-delivery systems allow drugs to act in specific sites of molecular action, which reduces the required dose of the drug, reduces the toxicity, and increases its local concentration [1]. In combination with the encapsulation of compounds with therapeutic properties, this system increases the drug permeability in tissues and its absorption or reduction in the frequency of administration of a given medication—all of this minimizes the occurrence of side effects [2]. Encapsulation is a physicochemical process involving the encapsulation of a target substance in a capsule made of a material of a different origin [3,4]. The covering material can be composed of many substances, from biological macromolecules forming liposomes and micelles [1,5], through biodegradable polymers [6], to metal nanoparticles [7]. Encapsulation is a technique used in many fields, which has been appreciated in tissue engineering as a technique that allows cells to be enclosed in a capsule made of a semipermeable membrane and implanted in an appropriate environment, allowing them to grow and divide, which contributes to tissue regeneration [8,9]. Encapsulation has also found application in the nutritional industry [3], as a drug carrier [1,2,6,10], or in gene therapies based on phage therapies [11]. Capsules improve the transport of drugs to the appropriate tissues, thanks to the increased absorption of the drug [2,12], and they minimize the side effects of medicinal substances, as well as sustain an extended release time of the drug into the body [2]. Additionally, the use of drug encapsulation and transport to a given tissue or target site of action allows for a rise in the local concentration of the drug while using a lower dose [1]. Drug capsules can be targeted at specific tissues using the basic tools of targeted therapy—monoclonal antibodies. There are modified forms of classic antibodies, including scFv (single-chain fragment variable), that have been used in modern therapeutic and diagnostic methods. ScFv is a fragment of a monoclonal antibody, consisting of the variable part of the heavy (V_H_) and light chains (V_L_), connected by a flexible peptide linker consisting of approximately 15 amino acids, most often glycine, serine, and/or lysine residues. The entire construct is approximately 30 kDa. ScFv has a similar binding specificity to the antibody from which it was produced, and at the same time, its immunogenicity is low [13,14,15]. Due to its small size and high antigen specificity, it has an increased ability to penetrate tissues, which is used in modern methods of treatment [13,14]. ScFv is used as a specific delivery molecule for substances such as radionuclides to specific tissues and cancer cells in radioimmunoimaging or radioimmunotherapy. They can also be used to deliver drugs, immunotoxins, cytotoxic proteins, or therapeutic genes in gene therapies [14,16,17,18] and in therapies for neurodegenerative diseases such as Alzheimer’s disease or Huntington’s disease [19,20]. Antibodies can also be used for coating metal nanoparticles, such as being made of gold or containing cysteine or histidine in the connecting region [21], and used as so-called target ligands for the controlled delivery of active substances to selected, well-defined target regions, thanks to the recognition of specific epitopes on receptors and their homo- and heteromers. Properly designed and prepared nanocapsules can have very low immunogenicity, which, in combination with the highly selective binding of antibodies, offers great therapeutic potential [22,23]. An additional advantage of scFv is that, due to the developing technique of genetic engineering and cloning of genes encoding antibodies, scFv can be cloned and expressed in various systems and cells, such as bacterial, eukaryotic (mammalian and yeast cells), insect [24,25,26,27,28], and plant cells [14,17,29]. One of the most popular and widely used methods of scFv selection is phage display [14,30]. The scFv antibody could potentially be used as a targeting ligand for drugs acting in the central nervous system (CNS); however, the passage of drugs through the blood–brain barrier (BBB) is extremely difficult due to the high specificity of the substances permeated, so that only 2% of small-molecule drugs are able to pass through. On the contrary, macromolecular drugs cannot pass through the BBB [31]. The difficulty in transport within the brain is influenced by two barrier connections. In addition to the BBB, which is mainly made up of the endothelium of cerebral capillaries (approximately 200 nm thick), there is also the blood-cerebrospinal fluid barrier (BCSFB) composed of the epithelium of the choroid plexuses. No molecules above 500 Da can pass through such a tight barrier; therefore, the astrocyte foot is attached to endothelial cells, taking carefully selected substances from them [32]. However, there is an inspiring example of the means to cross the BBB with the use of an antibody. The heavy chain end of a chimeric monoclonal antibody (MAb) against the human insulin receptor (HIR) has been humanized [33], and a significant penetration of the BBB, particularly efficient into grey matter [34], has been achieved following intravenous administration of the drug to *Rhesus macaque* individuals. Targeted therapy with the use of antibodies is a dynamically developing field of pharmacy, due to more specific drug targeting and delivery. Moreover, it allows for reduced toxicity while increasing the therapeutic effect [35]. It may become very useful in the treatment of schizophrenia, such as with the use of clozapine, an antipsychotic drug, which is an effective therapeutic that—at the same time—induces many undesirable side effects not only in the extrapyramidal system, but may also cause agranulocytosis [36]. Heterodimers of the dopamine D_2_ and serotonin 5-HT_1A_ receptors are important in the mechanism of action of clozapine and other novel antipsychotic drugs [37,38]. Therefore, antibodies of the scFv format, directed against such heterodimers, scFv_D2–5HT-1A_, might be useful in nanocarrier therapy. Heterodimers of the dopamine D_2_ and 5-HT_1A_ receptors occur in strictly defined areas of the CNS (e.g., in the prefrontal cortex) [39,40,41]; therefore, directing appropriate drugs to specific brain regions may reduce or even exclude many side effects, increasing the treatment efficacy [37].

Bearing in mind the above issues, in the present paper, we focused on characterizing the monoclonal antibody scFv_D2–5-HT1A_ as a targeting ligand dedicated to functionalization of the surface of nanocarriers for therapeutic compounds. The protein was previously obtained using phage display techniques from the phagemid library Tomlinson J [15]; its production was optimized in our previous study [42,43]. In the present study, we analyzed this protein and characterized it both biochemically and biologically. We show here the comprehensive characterization of the protein, including protein secondary structure analysis, estimation of immunogenicity and cytotoxicity, and receptor interactions, as well as its behavioral effect in mice. Finally, we conjugated nanogold particles with the scFv_D2–5-HT1A_ antibody. The nanoparticles are a model for nanocarriers, as they are able to cross the BBB [44,45] and are able to bind to the surface of a variety of biomolecules, such as proteins, which can modulate their activity [46].

## 2. Materials and Methods

### 2.1. scFv Production

The bacterial expression strain *Escherichia coli* Rosetta Blue (Novagen, Pruszkow, Poland) was transformed with a vector containing the scFv sequence, obtained previously from the phage-displayed scFv library Tomlinson I+J (Geneservice, Cambridge, UK) [15,42,43]. A single colony was grown in 5 mL of LB medium containing kanamycin (30 μg/mL) and chloramphenicol (34 μg/mL) at 37 °C with shaking for 8 h; then, the culture volume was changed to 50 mL. Following incubation overnight, this solution was used to inoculate 6 × 1 L of the same medium and cultured (37 °C, 180 RPM) until OD_600_ ≈ 0.6 was reached. Then, IPTG was added to a final concentration of 1 mM. After induction, the cells were cultured for 18 h at 28 °C (220 RPM); then, bacterial cells were centrifuged (5000× *g*, 10 min, 4 °C) and frozen at −80 °C.

Protein purification: The scFv sequence contained a periplasm export signal and thus could have been isolated from a bacterial periplasm. A bacterial pellet obtained from 1 L was suspended in 20 mL of buffer (50 mM Tris, 1 mM EDTA, 20% saccharose, pH 8.0, ice-cold) with protease inhibitor cocktail (Sigma-Aldrich, Poznan, Poland). The suspension was incubated with gentle rotation for 15 min at 4 °C and then centrifuged (30,000× *g*, 10 min, 4 °C). The supernatant was collected, and the pellet was resuspended in 20 mL of sterile water and then rotated and centrifuged likewise. Both supernatants were pooled and dialyzed to binding buffer (100 mM Na_2_HPO_4_, 150 mM NaCl, pH 7.2) at 4 °C. A column with protein L-agarose resin (Thermo Fisher Scientific, Warsaw, Poland) was equilibrated with binding buffer. The lysate was centrifuged (30,000× *g*, 30 min, 4 °C) and loaded onto the column in a loop for 3 h. After this time, the resin was rinsed with binding buffer. Elution was conducted with a buffer consisting of 0.1 M glycine pH 2.5, RT. The eluted fractions were immediately mixed with 10% of buffer 1 M Tris, pH 8.25, ice-cold. The content of the scFv protein was judged by SDS-PAGE and Coomassie staining. The fractions that contained the protein of interest were pooled and dialyzed to 100 mM MOPS buffer at 4 °C and frozen at −80 °C.

### 2.2. Protein Stability

The protein solution (100 μg/mL) was scanned on the JASCO-715 polarimeter at room temperature. The spectra were run from 200 to 250 nm in a 0.02 mm thick cuvette. Spectra were recorded three times and averaged, discarding points where the photomultiplier voltage was over 600 V. The measurements were performed for 10 mM MOPS buffer and the scFv_D2–5-HT1A_ protein in 10 mM MOPS buffer. The spectra were analyzed using the online applications K2D2 [47], K2D3 [48], and BeStSel [49,50].

The protein solution (100 μg/mL) was analyzed for denaturation temperature in a 1 mm thick cuvette. The wavelength was 279 mm with heating of 25–80 °C and a speed of 1 °C/min with an analysis step of 0.2 °C.

Additionally, the protein stability in the MEM + 10% FBS medium at 37 °C was tested with CD scanning.

The medium was mixed either with PBS (control) or a scFv_D2–5-HT1A_ protein solution in PBS (test) in a ratio of 3:1. The final protein concentration in the test group was 26.75 μg/mL. The incubation times varied from 0 to 180 min, with measurements performed every 15 min during the first hour and every 30 min thereafter. CD scans were performed at 200–250 nm. The value obtained for the control group was subtracted from that of the test group.

### 2.3. Cell Culture

Human embryonic kidney cells (HEK 293) were cultured in MEM with 10% fetal bovine serum (FBS) (Sigma-Aldrich); mouse murine macrophages cells (RAW 264.7) were cultured in DMEM supplemented with 1% L-glutamine, high glucose, and 10% FBS (Sigma-Aldrich); rat pheochromocytoma cells (PC12) were cultured in RPMI-1640 with 10% FBS (Sigma-Aldrich). The immortalized human cerebral microvascular endothelial cells, a D3 clone (hCMEC/D3) kindly provided by Prof. Babette Weksler, were grown in EBM-2 (endothelial cell basal medium) supplemented with components obtained from the manufacturer: Human epidermal growth factor (hEGF), vascular endothelial growth factor (VEGF), R3-insulin-like growth factor-1 (R3-IGF-1), ascorbic acid, hydrocortisone, human fibroblast growth factor-beta (hFGF-β), heparin, gentamicin/amphotericin-B (GA), and 10% FBS (Lonza, Basel, Switzerland). To provide optimal growth conditions, the cells were seeded on tissue culture plastics coated with rat collagen (Sigma-Aldrich) (final protein concentration of 150 g/mL). hCMEC/D3 cells were used in passages 26–34 because only under this condition do cells show BBB features. After the cell culture conditions were established, the expression of endothelial cell marker adhesion, tight junction proteins, and ABC and SLC transporters (characteristic for cells engaged in forming the BBB) were checked, as described in our previous publication [51]. Moreover, TEER (transepithelial/transendothelial electrical resistance) measurements were performed using a Milicell ERS-2 Volt-Ohm Meter (Millipore Poznan, Poland)) and STX01 chopstick electrode (Millipore). hCMEC/D3 cells were seeded onto inserts with a pore size of 1.0 µm (Falcon, Becton, Dickinson and Company, San Jose, CA, USA) in concentrations of 100,000/well. The cells were cultured for five days before measurement. The measurements were regularly repeated to ensure proper barrier properties. The final TEER was calculated by subtracting the resistance of the insert without cells from the resistance of the insert with hCMEC/D3 cells. The numbers were then multiplied by the growth area.

All of the cell lines were cultured at 37 °C in a humidified incubator with a 5% CO_2_ atmosphere.

### 2.4. Cytotoxicity Tests

MTT: RAW 264.7 cells (30,000/well) and HEK 293 cells (100,000/well) were seeded on 96-well plates. Following stabilization overnight, the cells were incubated with scFv_D2–5-HT1A_ in DMEM/MEM + 10% FBS, with a protein concentration of 32.75–0.002 μg/mL. After 4 or 24 h, the medium with protein was changed to medium with 0.5% MTT without FBS. After another 4 h, the medium was removed, and formazan was dissolved in DMSO. The plate was read in a TECAN Infinitive M200 Pro plate reader (TECAN) at 570 nm.

Cell titer: RAW 264.7 cells (15,000/well), HEK 293 cells (50,000/well), and PC12 cells (50,000/well) were seeded onto a 96-well plate half area and cultured overnight at 37 °C. Then, the media were changed for the solution containing scFv_D2–5-HT1A_, with a protein concentration of 32.75–0.002 μg/mL. After 4 or 24 h of incubation, the media were changed for media with 20% cell titer blue. After 3–4 h, the fluorescence was measured with excitation at 560 nm and emission at 590 nm using a TECAN Infinitive M200 Pro plate reader (TECAN).

LDH: RAW 264.7 cells (30,000/well) and HEK 293 cells (100,000/well) were seeded on 96-well plates. Following stabilization overnight, the cells were incubated with scFv_D2–5-HT1A_ in DMEM/MEM + 10% FBS, with a protein concentration of 32.75–0.002 μg/mL. After 4 h of incubation, the plates were centrifuged (1000 RPM, 7–10 min), and a test was conducted according to an LDH Cytotoxicity Detection Kit (Takara Bio). The absorbance at 490 nm was measured with reference at 610 nm using a TECAN Infinitive M200 Pro plate reader (TECAN).

All the tests were conducted with PBS buffer as the control group since the protein was prepared in this buffer.

### 2.5. Second Messengers

cAMP: The test was conducted according to a cAMP Gs Dynamic Kit (CisBio, Paris, France). The fluorescence was measured (TECAN Infinitive M200 Pro plate reader (TECAN, Männedorf, Switzerland) with excitation at 340 nm and emission at 620 and 665 nm for two concentrations of the scFv_D2–5-HT1A_ protein: 33 μg/mL and 16.5 μg/mL for the four cell lines: HEK 293 and HEK 293 expressing either the D_2_ or the 5-HT_1A_ receptor, or both of them.

IP_1_: The test was conducted under the same conditions as the cAMP test, using an IP-One Gq Kit (CisBio).

### 2.6. Cytokine Production

RNA isolation: RAW 264.7 cells were seeded onto 6-well plates (300,000/well) and cultured for 24 h. Then, they were incubated for 4 h with the scFv_D2–5-HT1A_ protein (concentration of 32.92–0.004 μg/mL), with each concentration being a double dilution of the previous one on the scale. After incubation, the cells were lysed using Trizol solution, and RNA was isolated with the phenol–chloroform method. Then, cDNA was obtained using reverse transcriptase (Thermo Fisher Scientific).

qPCR: The expression levels of IL-6, TNF, iNOS, and iκB were tested using a qPCR Master Mix (Thermo Fisher Scientific) with GAPDH as a reference gene on Illumina Eco Termocycler (Illumina, Cambridge, UK).

Primer sequencences are shown below in Table 1.

### 2.7. Nanogold Coupling

scFv_D2–5-HT1A_ was coupled with nanogold particles using an 80 nm InnovaCoat GOLD Conjugation Kit (Interchim, Montluçon Cedex, France). The protein solution was diluted to 50 µg/mL using an antibody diluent solution and 12 µL of such diluted scFv_D2–5-HT1A_ mixed with 42 µL of reaction buffer. Then, 45 µL of the mixture was added to gold particles and incubated for 15 min at room temperature. The reaction was stopped using 5 µL of the quencher solution. The prepared product was then added to hCMEC/D3 cells using 5 µL per well.

For ELISA tests, the nanogold conjugates were also washed to remove any unbound antibodies. Next, 50 µL of the conjugate was washed with 500 µL of the quencher solution diluted to 1:10 and then centrifuged (9000× *g*, 10 min, room temperature).

### 2.8. Transmission Electron Microscopy Imagining of the Nanogold Particles

Samples of scFv_D2–5-HT1A_ protein coupled to nanogold particles, as well as free nanogold particles, were collected on 300 mesh grids made from copper; additionally, the latter was covered with formvar film. For observation, the JEOL JEM 2100HT electron microscope (Jeol Ltd., Tokyo, Japan) was used at accelerating voltage 80 kV. Images were taken by using 4k × 4k camera (TVIPS) equipped with EMMENU software ver. 4.0.9.87.

### 2.9. ELISA

The protein of interest was coupled with nanogold particles (Innova) according to the manufacturer’s protocol [52]. The HEK 293 cells, as well as the HEK 293 cells expressing dopamine, the D_2_ or serotonin 5-HT_1A_ receptor, or both, were suspended in medium and placed in a cone-shaped 96-well plate (150,000 cells/well). They were then incubated with the gold-coupled scFv_D2–5-HT1A_ protein for an hour on ice. After rinsing the cells with cold PBS (four times), a mixture of 0.05% protein L-HRP conjugate with 0.5% BSA was added to each well. Again, the cells were rinsed with cold PBS. After adding the TMB reagent, the reaction was developed and stopped with 1 M HCl. The resulting absorbance was measured at 450 nm.

### 2.10. The Blood–Brain Barrier Model Transition

The hCMEC/D3 cells were seeded (100,000 cells/well) onto a 12-well plate with collagen-coated inserts with 1.0 µm pores. The cells were cultured for five days (EGF-2 medium (Lonza) + 10% FBS). After 2, 4, and 24 h, the flow-through media were collected, and absorbance at 550 nm was measured using a TECAN Infinitive M200 Pro plate reader (TECAN). We used medium without nanoparticles as a negative control, while as a positive control, medium solution with 5 µL (100%) of the nanogold conjugate was used. A passive crossing test using medium from an insert without cells was also undertaken.

### 2.11. Animals and scFv_D2–5-HT1A_ Antibody Administration

The 3-month males of C57BL/6 mice were purchased from Charles River Laboratories (Germany). The animals were divided into the 5 groups (n = 8–14) and subjected to handling for 7 days. They had free access to food and water and were kept at a constant room temperature (24 °C) under a 12 h light/dark cycle. Afterward, the mice were administrated i.p. 100 µL of phosphate buffer, scFv_D2–5-HT1A_ antibody (1 mg/mL), gold nanoparticles, gold nanoparticles with the scFv_D2–5-HT1A_ antibody, and clozapine (0.3 mg/kg). The locomotor activity of animals was measured starting at 30 min after administration for 60 min.

All procedures were carried out in accordance with the decision of the II Local Bioethics Commission (198/2021).

### 2.12. Locomotor Activity

The locomotor activity of mice was measured in OPTO-M3 cages (Columbus Instruments, Columbus, OH, USA) connected to a compatible PC. Each mouse was placed individually in the transparent acrylic plastic chamber (13 cm × 23 cm × 15 cm) surrounded by an array of photocell beams for 60 min. The horizontal activity was measured as disruption of the photobeams and was defined as ambulation scores. The results were counted every 10 min.

### 2.13. Statistical Analysis

The results obtained in this study came from different experiments repeated a few times. The data are presented as mean ± standard error of the mean (SEM). The statistical significance was evaluated using Student’s *t*-test and Mann–Whitney *U*–test (* *p* < 0.05, ** *p* < 0.01, and *** *p* < 0.001). Significance was estimated by an ANOVA test.

The statistical analysis of the animal study was performed with GraphPad Prism 8.4.3. One-way ANOVA (followed by Tukey’s post-hoc test) was used to determine the statistical significance of differences between the groups. The *p*-values ≤ 0.05 were considered statistically significant

## 3. Results and Discussion

### 3.1. Protein Production

The sequence encoding the scFv_D2–5-HT1A_ antibody was obtained using the phage display technique from the phagemid library Tomlinson J, as described previously [15,42,43]. The scFv_D2–5-HT1A_ antibody was obtained by isolation from the bacterial periplasm. Under regular conditions, the bacterial cytoplasm is not a perfectly suitable environment for the expression of eukaryotic proteins due to its high reduction potential, acidic pH, and high protease activity [53]. Usually, there is a need to use an appropriate bacterial strain, with an expression of chaperones and thioredoxin reductase, which allows the formation of disulfide bonds [54]. However, there is still a high probability of overexpression and the formation of inclusion bodies. Thus, the scFv_D2–5-HT1A_ protein was enriched with the periplasm export signal, as in the periplasm, there is no such strong reducing environment, and there are much fewer proteases, which allows the protein to fold properly and avoid degradation.

The scFv_D2–5-HT1A_ protein with the periplasm export signal was expressed in *E. coli* Rosetta Blue strain and purified by affinity chromatography, using resin with protein L, which is a perfect tool to isolate recombinant antibodies from bacteria lysates. It binds to structural antibody motives only [55] and thus allows one to achieve both high purity and high functionality for the eluate, as it does not bind denatured proteins.

As can be seen in the gel pictures (Figure 1), the eluted fractions contained almost exclusively the protein of interest (with a molecular mass of approximately 27 kDa). The only impurity was a darker cloud slightly under the protein image, around 18 kDa. During purification, the elution step was performed with glycine buffer at 2.5 pH. This is quite a harsh environment for the protein, and even if the solution is neutralized as soon as the eluate is out of the column, protein degradation is likely to occur. Another possibility is that the eluted fragments contained the protein that was degraded during isolation, and those fragments were fragments of scFv_D2–5-HT1A_ with structural motives that interact with the column resin. Nevertheless, the quality and purification of the scFv_D2–5-HT1A_ protein were high, and thus, it can be assumed that this method of isolation is efficient and affords a high yield (there was, at most, the residual part of the protein in the flow-through).

### 3.2. Protein Stability

Proper functioning of the protein depends greatly on its structure, so it was necessary to check the CD spectra of the purified scFv_D2–5-HT1A_. First, it was checked if the protein was properly folded; the results are presented in Figure 2A.

The spectra were then analyzed using the web applications K2D2 [47], K2D3 [48], and BeStSel [49,50] to obtain the percentage of structural motives in the protein.

The results of the analysis (Figure 2A and Table 2) are consistent with the available data for the secondary structure of immunoglobulins and scFvs format of antibodies [56], which indicates that the purified scFv_D2–5-HT1A_ protein was properly folded. The experimental data are also consistent with the amino acid sequence analysis for immunoproteins (BLAST, Jpred, PredictProtein).

Next, the denaturation temperature of the scFv_D2–5-HT1A_ protein was analyzed with a temperature scan at 279 nm (Figure 2B). A sigmoid function was fitted to the plot and differentiated. The maximum value of the differential function was read.

The denaturation temperature turned out to be 55.47 °C. The usual temperature of denaturation of similar immunoglobins is around 61 °C [56,57], so our results remain in agreement with other studies.

To check if scFv_D2–5-HT1A_ is stable under the experimental conditions, a solution of the protein in MEM + 10% FBS at 37 °C (final concentration of 25.75 µg/mL) was monitored for 3 h. The results are shown in Figure 3.

After the initial fall (0–15 min), the signal remained rather constant, with small fluctuations, which indicates that the obtained scFv_D2–5-HT1A_ protein is stable under the experimental conditions for at least 3 h.

Analogous analysis was demonstrated by Lee et al. [58], who obtained a similar thermal stability of 3D8 scFv, a catalytic antibody with nucleic acid-binding and -hydrolyzing activities. The working ability of the antibody decreased when incubated at more than 50 °C for over 2 h. 3D8 scFv has also showed a slight decrease when incubated for longer periods of time. Although the initial fall of scFv_D2–5-HT1A_ was larger, the protein stopped decreasing, while 3D8 scFv continued. Moreover, the results obtained by Montoliu-Gaya et al. [59] indicated that scFv-h3D6 (anti-amyloid β) has a similar but slightly higher denaturation point (closer to 60 °C). The results of Sarker et al. [60], who studied scFv constructed from an anti-fusion loop dengue E53 Fab antibody, showed that compared to the protein, scFv_D2–5-HT1A_ has a more defined secondary structure, as in the “double wavelength” chart, scFv_D2–5-HT1A_ is closer to the native group than to the pre-molten globule.

### 3.3. Transmission Electron Microscopy of Nanogold Particles

To ensure that the properties of nanogold particles are consistent with the manufacturer’s description, the nanogold particles, both coupled with scFv_D2–5-HT1A_ protein and free, were imaged with transmission electron microscope.

As can be seen in Figure 4, the nanogold particles are indeed sized around 80 nm in diameter, and both the free and coupled nanogold particles are indistinctive even on such a scale. The differences are, however, noticeable via cellular and animal experiments, as we present below.

### 3.4. Cytotoxicity

The purified scFv_D2–5-HT1A_ was properly folded, as shown above. The next important issue, which constitutes the key point in designing novel drug carriers, concerns the potential cytotoxic effects of the carrier and hence the targeting ligand—the scFv_D2–5-HT1A_ antibody.

The scFv_D2–5-HT1A_ protein potential cytotoxicity was tested on HEK 293 and PC12, as well as on RAW 264.7. On the other hand, the effects of scFv_D2–5-HT1A_ protein on secondary messenger production were studied for HEK 293 cells, which are considered a proper cell line for neuronal receptor expression, and thus simpler for culturing a neuronal model. PC12 cells, despite being pheochromocytoma cells, show properties of mature dopaminergic neurons [61] to a certain extent because they can release dopamine [62], thus serving as a good model to study neurotoxicity [61,62]. RAW 264.7 cells are a mouse macrophage line (Abelson murine leukemia virus-induced tumor), which can be used for an initial immunogenicity test, as they are able to produce cytokines. They are considered an appropriate model of macrophages [63] and are widely used in inflammation research [64,65].

Such analysis is crucial to ensure proper results in further examinations. The measurements were conducted at two time points, 4 or 24 h, in the presence of scFv_D2–5-HT1A_ in the incubation medium, depending on the test. To be sure that the cell viability experiments are valuable, we decided to use various tests (cell titer, MTT, and LDH). The studies showed that the cell titer blue and MTT assays can indicate slightly different properties of the viability and thus should be treated as complementary [66]. These tests were conducted with a control group, where only buffer (PBS) was added.

For higher consistency of the experiment, another assay was performed—LDH measurements. This test records the activity of lactate dehydrogenase, an enzyme that, under normal conditions, resides inside the cell. However, after cell death, LDH is released into the medium, where it can be detected [66]. In this kind of experiment, the control is considered to undergo 0% cell mortality. As a control group, a few wells are treated with 1% Triton X-100 to achieve 100% mortality, and based on these, the cell viability is calculated.

The results indicate that the broad protein concentration range (0.002–33 µg/mL) did not induce any cytotoxic effect in the cell lines (HEK 293, PC12, RAW 264.7) subjected to all the tests. In Figure 5, representative data of the cell viability experiments for RAW 264.7 cells are shown. None of the results indicated a significant decrease in viability, and there was no significant difference between the control and experimental groups. The factor with the greatest influence on results was the medium dilution.

Cytotoxicity tests are crucial in examining the protein of interest. Our aim was to prepare the scFv_D2–5-HT1A_ most appropriate for future targeted therapies, and thus both the protein and whole construct must not have cytotoxic properties. There is always a possibility that a nanocarrier with a targeting ligand has toxic effects—if such a thing happens, the therapeutical abilities would be heavily disturbed.

Therefore, cytotoxicity tests are important. Surprisingly, they are not often conducted, as the literature indicates [67,68,69]. We are strongly convinced that a complex and comprehensive analysis of a potential therapeutic’s cytotoxicity is crucial from a perspective of novel drug design and delivery.

### 3.5. Cytokine Production

As the scFv_D2–5-HT1A_ protein was designed to be used as a nanocarrier surface-functionalizing molecule, it should not present immunogenic properties. We approached this issue by measuring the alterations in the levels of mRNA-encoding cytokines [70]. We used qPCR with the GAPDH gene as a reference gene. Experiments were conducted on RAW 264.7 cells, which are mouse macrophage cells. Cytokine production tests are considered to be reliable in immunogenicity estimation [71].

The results shown in Figure 6 were normalized to control cells, which were incubated in the presence of PBS.

Each of these proteins plays a significant role in the immunological response. Interleukin 6 (IL-6) is a small (21 kDa) pro-inflammatory molecule that mediates the immune response [72,73]. It is also a stimulator of B-cells [73]. Meanwhile, TNF is both a mediator and regulator of the immune response [74], but it is also considered a cell survival and proliferation agent. iNOS is one of three nitric oxide synthases, next to neuronal and epithelial isoforms, all of which are able to produce nitric oxide, albeit in different kinds of cells and under different conditions [75]. iNOS is activated in response to cytokines, usually under stress, but also during inflammation [76]. The last of the tested proteins was IκB, which is an inhibitor of NFκB (nuclear factor kappa-light-chain-enhancer of activated B cells) [77], which acts as an apoptosis, cell proliferation, and inflammatory modulator [78]. Under normal conditions, NFκB resides in the cytoplasm, where it is bound by the IκB and is thus unable to translocate to the nucleus. NFκB can be activated through the TNF response, and then the inhibitor is proteolyzed [79]. All of these cytokines provide complex information about the possible inflammatory condition of RAW 264.7 cells upon incubation with the scFv_D2–5-HT1A_ antibody.

The obtained results showed weak to no immunogenicity of the scFv_D2–5-HT1A_ protein—most of the tested protein concentrations did not induce an increase in IL-6, TNF, or iNOS. The only protein that was elevated in the majority of the tests was IκB, which acts as a suppressor of the immune response [80], further proving that the pro-inflammatory potential of the scFv_D2–5-HT1A_ protein is low to non-existent. These results allow the conclusion that the scFv_D2–5-HT1A_ protein—which aims to functionalize the nanocarrier surface for medical use—does not possess any immunogenic properties, which is advantageous, especially in the context of scarce literature [81]. Similar to cytotoxicity tests, immunogenicity analysis data in the literature are not often encountered. Such tests were performed only in the aforementioned studies regarding vaccines [82] and chimeric T-cells [83], where immunogenic activation was desired. However, few—if any—scFvs used for other purposes have been tested in this fashion.

### 3.6. Second Messengers

All of the analyses described above indicate that the scFv_D2–5-HT1A_ protein was properly folded and stable, with neither cytotoxic nor high immunogenicity. However, the next step was to study its functional influence on the dopamine D_2_ and serotonin 5-HT_1A_ receptors and their heteromer. To this end, experiments based on the FRET (Förster resonance energy transfer) phenomenon were performed to estimate the second messenger levels (IP1 and cAMP) formed upon the incubation of HEK 293 cells expressing the desired receptors (D_2_R, 5-HT_1A_R, or both) with the scFv_D2–5-HT1A_ antibody (Figure 7).

There was no significant difference in the production of second messengers between the cells expressing one of the studied receptors or both of them (*p* = 0.54); the signal levels were also similar upon the incubation of the cells with both concentrations of the scFv_D2–5-HT1A_ antibody. However, those three cell lines were statistically different from the non-transfected cells (*p* < 0.05) as far as cAMP was concerned. It is important to note that the cells with co-expression of both receptors showed a smaller signal in the cAMP assay than the sum of the two consistent receptors. The non-additive signal level can indicate crosstalk between the receptors or their signaling pathways. GPCRs are able to interact with other receptors [84]. Dopamine receptors are known for their ability to form heteromers, which can alter their functionality [85] and lead to a unique pharmacological profile [86,87], which changes their functional properties [88]. The IP1 test did not show any difference between the tested cell lines, which indicates that the scFv_D2–5-HT1A_ protein did not activate the phospholipase C pathway.

As mentioned throughout the manuscript, scFv_D2–5-HT1A_ is meant to be used as a targeting agent to functionalize nanocarriers with antipsychotic drugs. In one of our publications, we compared a few drugs [88] that can act as a reference with the results obtained in the present paper. In the context of cAMP, scFv_D2–5-HT1A_ had a suppressing effect on the tested receptors, similar to lurasidone or aripiprazole. For the IP-1 pathway, the protein had almost no influence, which can be compared to lurasidone or clozapine. Significantly, the protein can influence the receptors. This property is crucial, as scFv_D2–5-HT1A_ does not only play the role of a targeting ligand, but it can also act upon the receptor proteins themselves, thus becoming the active part of the whole complex, thereby enhancing its properties.

It has to be realized that HEK 293 cells express endogenous dopamine D_2_ receptors, so scFv_D2–5-HT1A_ can also influence the control cells.

### 3.7. ELISA

The process of functionalization of a nanocarrier’s surface can enhance its properties and improve its specificity [89]. It is the selectivity toward the heterodimer of the dopamine D_2_ and serotonin 5-HT_1A_ receptors that is the key feature, enabling minimization of the side effects of the drugs they carry.

The most frequently used test to estimate the parameters of surface molecules on nanocarriers is ELISA. This assay can be modified to assess different kinds of proteins—antigens in a solution, antibodies in the blood, or antigen–antibody affinity. Using this test, the properties of the scFv_D2–5-HT1A_ antibody obtained in the present study were estimated, and their two forms were compared, i.e., coupled with nanocarriers and in a solution. The results are shown in Figure 8.

The scFv_D2–5-HT1A_ protein coupled with nanocarrier binds better (*p*-value ≤ 0.05) with cells than free protein. In our previous preliminary study, we showed that it binds stronger with cells co-expressing both receptors than with cells expressing only one kind of receptor [43], and here, we added important information concerning the obtained scFv_D2–5-HT1A_ antibody—it shows better interaction properties with receptors when it is bound onto the nanocarrier surface. Since this protein involves nanocarrier surface functionalization, one may expect that it enhances the surface, ensuring selectivity and affinity to targeted cells. Gold nanocarriers are non-toxic [90,91], non-immunogenic [46], and possess the ability to cross the BBB [45]. They can be safely treated as a valid model of more complex nanocarriers—they have a surface with the ability to be functionalized [46] and size variations similar to such nanostructures [92]. Free scFvs have also been proven to cross the blood–brain barrier [93]; however, as shown in Figure 8, there was much stronger binding with scFv_D2–5-HT1A_ on nanogold particles than in a solution.

### 3.8. The Blood–Brain Barrier Model Transition

Previous experiments have shown that the obtained scFv_D2–5-HT1A_ protein is functional and non-immunogenic. Since it is meant to be used to address the molecules on the surface of nanocarriers, which are supposed to cross the BBB, in the next stage of experiments, we decided to test this property.

hCMEC/D3 are widely used as a BBB model for in vitro examination [94]. They have the ability to form a very strict monolayer, which is resistant to electrical current, as well as to the penetration of certain agents and chemicals [94,95]. We have wide-ranging experience in working with these cells, as has been described in our previous publication [51]. The conditions of the experiments were strictly defined and checked in order to be sure that the BBB properties were retained each time. TEER measurement is a standard operation while working with hCMEC/D3 cells in our laboratory. In the present study, we measured the TEER in order to be sure that the experiment was conducted properly. The obtained results ranged between 6 and10 Ω·cm^2^ and are consistent with the results obtained by Eigenmann et al. [95] and Biemans et al. [96].

hCMEC/D3 cells are also able to initiate transcytosis, which can be observed under specific conditions—such as when the cells are placed in insert wells that are filled and submerged into media. In such a way, there is the possibility to observe and test both their flow-through and supernatant. It is important to note that nanogold particles, which are detected by this assay, are highly immune to degradation [52].

The obtained results, presented in Figure 9, indicate that there is a slight reverse correlation between the incubation time and percentage of nanoparticles that crossed the barrier—the more time passed, the fewer particles were in the solution. This could be connected with efflux properties of the BBB, in that unwanted substances are able to cross back to the blood [97,98]. There is a possibility that such a mechanism took place in this experiment.

### 3.9. Locomotor Activity

We have presented that scFv_D2–5-HT1A_ protein does not show cytotoxicity and does not trigger the cytokine response. We also proved that it can influence secondary messenger levels and is able to cross the model of nanovesicles. In the next stage of our study, we performed a preliminary experiment using animals to obtain the information concerning the behavioral effect of the scFv_D2–5-HT1A_ protein (both in solution as well as bound to the nanogold particles) in comparison to clozapine, well established antipsychotic drug, acting—among others—on the same receptor heterodimers. The results of the performed analysis are shown in Figure 10.

Locomotor activity of mice was reduced by clozapine (0.3 mg/kg)—the drug used as reference in this study. This result remains in agreement with the previous study [99]. A similar effect was obtained in groups of animals receiving either scFv_D2–5-HT1A_ antibody or scFv_D2–5-HT1A_ antibody on gold nanoparticles.

## 4. Conclusions

The scFv_D2–5-HT1A_ protein’s ability to cross the blood–brain barrier on nanocarriers, taken with all of the properties described in the present study, including low immunogenicity, stability, and ability to affect the desired receptors, indicating that it is a promising agent that could play an important role in future therapies where the specific site of action of a given therapeutic plays an important role.

## Figures and Tables

**Figure 1 biomolecules-12-00749-f001:**
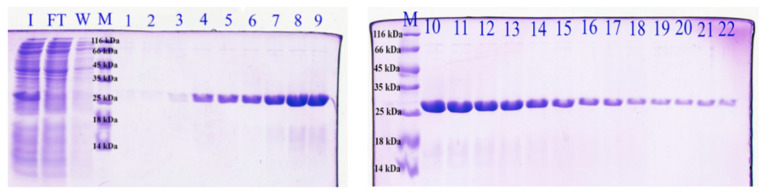
SDS-PAGE of scFvD2–5-HT1A. I, supernatant of the bacterial lysate; FT, column flow-through; W, column wash-out; M, weight marker; 1–22, number of fractions collected following affinity chromatography.

**Figure 2 biomolecules-12-00749-f002:**
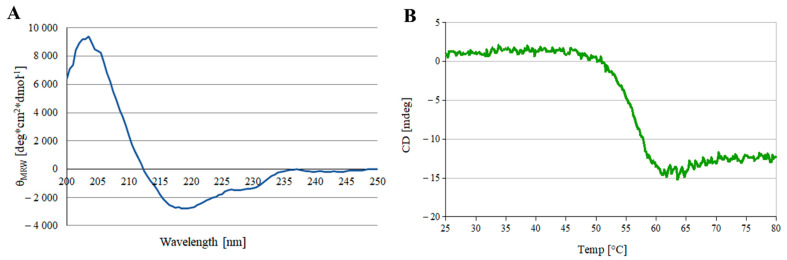
CD analysis of the scFv_D2–5-HT1A_ protein. (**A**) CD spectrum of the protein. The curve is the mean value of three separate scans with the buffer values subtracted. (**B**) Temperature scan at 297 nm used to access the denaturation temperature.

**Figure 3 biomolecules-12-00749-f003:**
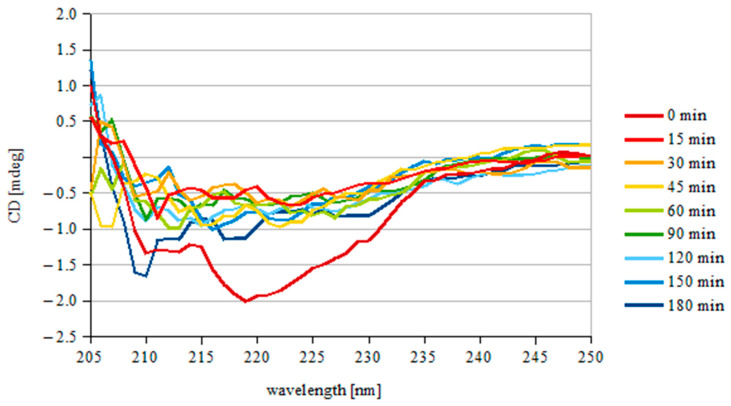
Protein stability under the experimental conditions. Each curve is the mean value of three separate scans. The scFv_D2–5-HT1A_ protein was incubated in MEM + 10% FBS for 15–180 min and then scanned. The values for the medium without scFvD_2_–5-HT1A were subtracted.

**Figure 4 biomolecules-12-00749-f004:**
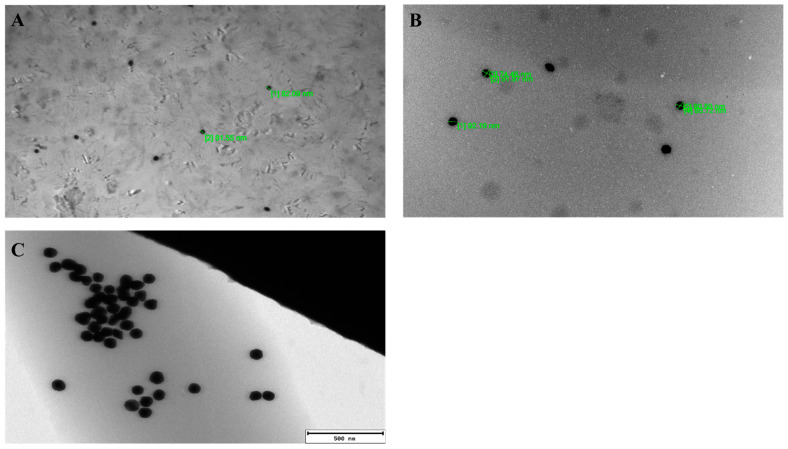
Transmission electron microscopy image of the nanogold particles. (**A**) Nanogold without protein. (**B**) Nanogold coated with scFv_D2–5-HT1A_ protein. (**C**) High-resolution picture of scFv-coated nanogold particles.

**Figure 5 biomolecules-12-00749-f005:**
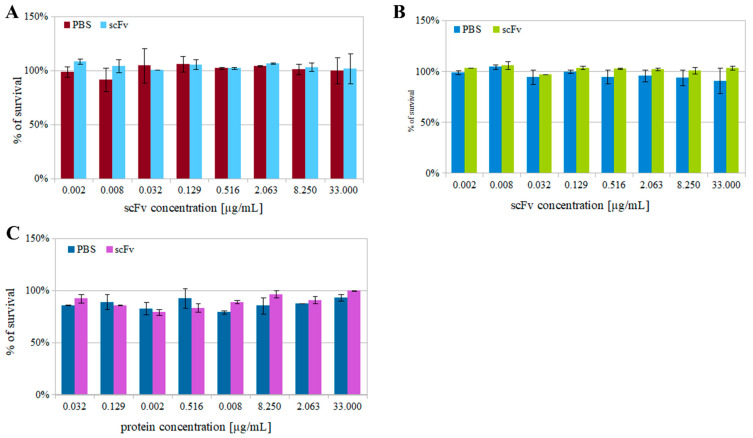
Cytotoxicity tests. Representative examples of RAW 294.7 cytotoxicity test. (**A**) MTT test on the RAW 294.7 cell line after 4 h of incubation. (**B**) Cell titer test on the RAW 264.7 cell line after 4 h of incubation. (**C**) LDH test on the RAW 264.7 cell line after 4 h of incubation. Errors were estimated using the standard deviation calculated from three separated repeats. PBS, phosphate buffer saline (negative control); scFv, scFvD_2_–5-HT1A in PBS.

**Figure 6 biomolecules-12-00749-f006:**
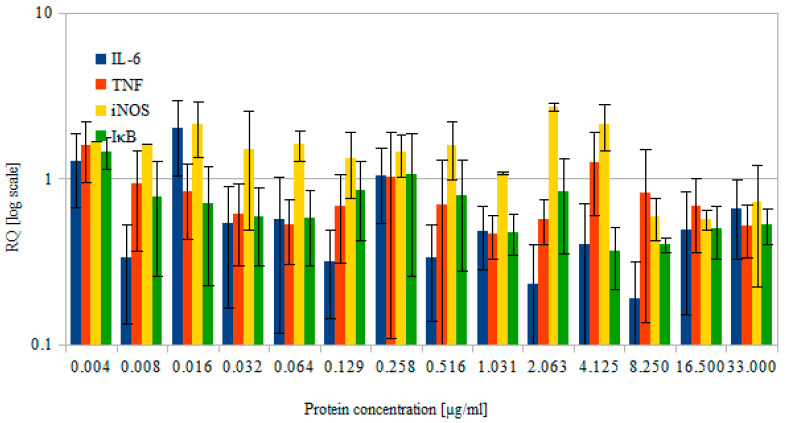
Relative quantity of selected cytokines produced by RAW 264.7 cells after 4 h of incubation with various concentrations of the protein of interest (scFvD_2_–5-Ht1A). Logarithmic scale. Errors were estimated using the standard deviation calculated from three separate experiments. IL-6, interleukin 6; TNF, tumor necrosis factor; iNOS, inducible nitric oxide synthase; IκB, inhibitor of the NFκB transcription factor.

**Figure 7 biomolecules-12-00749-f007:**
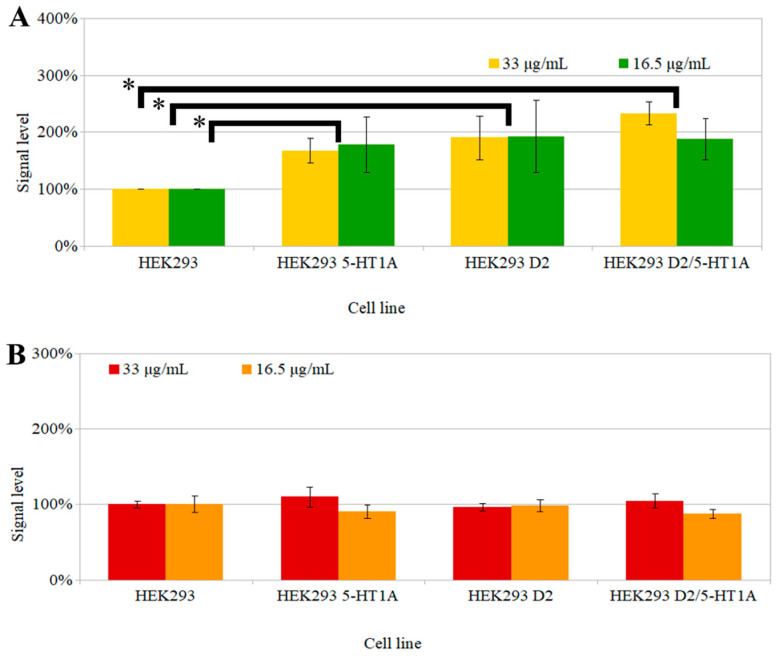
Homogeneous time-resolved fluorescence (HTRF) test results. On the X-axis are the cell lines used for the experiment. On the Y-axis is the signal, calculated by the ratio of the signal of the tested cells over the control (HEK293 cells). Two protein concentrations were used: 33 and 16.5 µg/mL. Errors were estimated using the standard deviation calculated from three separate experiments. (**A**) cAMP (cyclic adenosine monophosphate) test; due to competitive character of the test, a higher signal indicates a lower amount of cAMP in cells; * *p* < 0.05 vs. HEK 293 cells (control); (**B**) IP1 (inositol monophosphate), no significant differences between the cell lines were observed.

**Figure 8 biomolecules-12-00749-f008:**
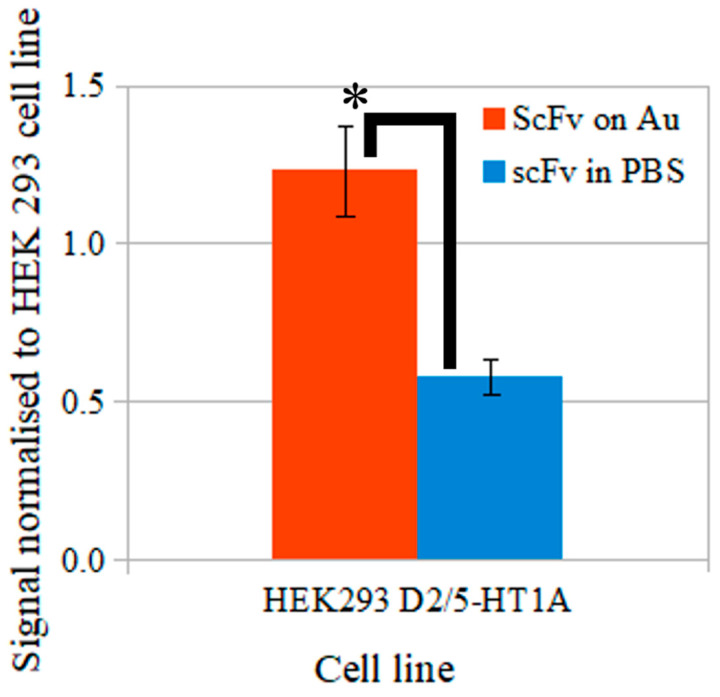
Binding of scFv antibody to heterodimers of dopamine D_2_ and serotonin 5-HT_1A_ receptors expressed in HEK293 cells. Assay was performed using ELISA, absorbance at 550 nm, the signal level was calculated by ratio of the signal in the tested cells over the control (HEK293 cells). ScFv on Au, antibodies coupled to nanogold particles; scFv in PBS, free antibodies; errors were estimated using the standard deviation calculated from three separate experiments. * *p*-value ≤ 0.05.

**Figure 9 biomolecules-12-00749-f009:**
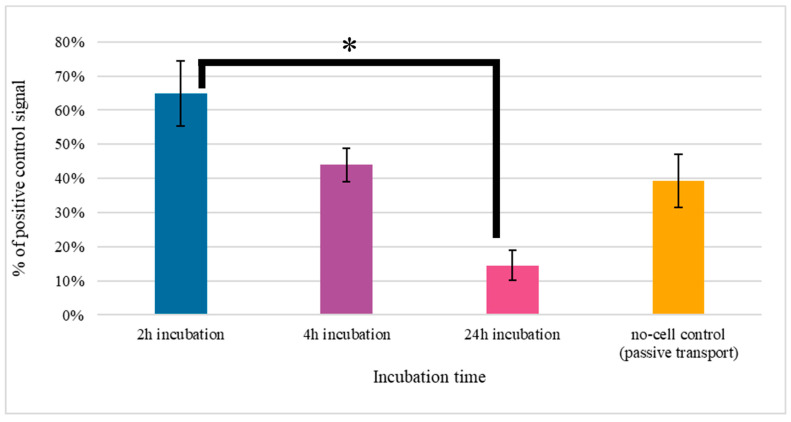
Transition experimental results obtained with hCMEC/D3 cells. Errors were estimated using the standard deviation calculated from repeats. Before the test, the integrity of the monolayer was checked, and the test was conducted only if the TEER factor indicated the existence of a monolayer. * *p*-value ≤ 0.05.

**Figure 10 biomolecules-12-00749-f010:**
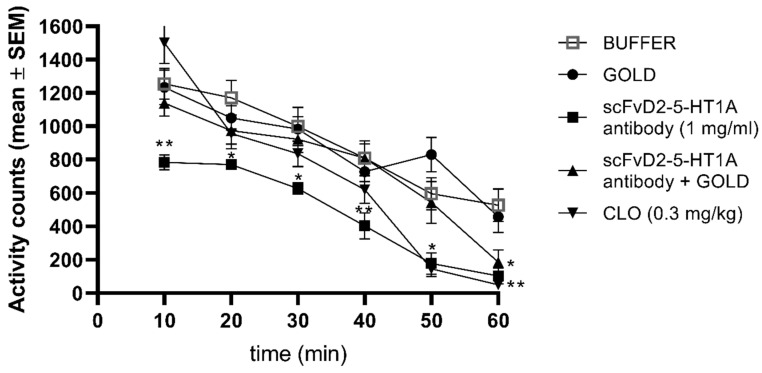
The locomotor activity of mice following i.p. administration of 100 µL of phosphate buffer (BUFFER), gold nanoparticles (GOLD), scFvD_2_–5-HT1A antibody (1 mg/mL); gold nanoparticles with the scFvD_2_–5-HT1A antibody and clozapine (0.3 mg/kg, CLO). Data are presented as mean ± SEM of activity counts over 60 min test in 10 min intervals and were analyzed using one-way ANOVA. The significant changes vs. control group (BUFFER) marked on the graph: ** *p* ≤ 0.01; * *p* ≤ 0.05; n = 8–14.

**Table 1 biomolecules-12-00749-t001:** qPCR primer sequences. IL-6, interleukin 6; TNF, tumor necrosis factor-alpha; iNOS, inducible nitric oxide synthase; iκB, inhibitor of nuclear factor κB.

Gene	Starter Forward (5′→3′)	Starter Reverse (5′→3′)
GAPDH	TCAACGGCACAGTCAAGG	ACTCCACGACATACTCAGC
IL-6	TTCTCTGGGAAATCGTGGAAA	TCAGAATTGCCATTGCACAAC
TNF	CCCTCACACTCAGATCATCTTCT	GCTACGACGTGGGCTACAG
iNO	TCCTACACCACACCAAAC	CTCCAATCTCTGCCTATCC
iκB	CTTGGTGACTTTGGGTGCTGAT	GCGAAACCAGGTCAGGATTC

**Table 2 biomolecules-12-00749-t002:** Structural motive comparison, calculated from online applications using the curve shown in Figure 2A.

Structural Motif	K2D2 and K2D3	BeStSel
Alpha helix	1.12%	1.70%
Beta sheet	36.30%	38.00%
Unstructured/other	61.95%	60.30%

## Data Availability

Department of Physical Biochemistry, Faculty of Biochemistry, Biophysics and Biotechnology, Jagiellonian University.

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
