# Peer review of "Dopamine D2 and Serotonin 5-HT1A Dimeric Receptor-Binding Monomeric Antibody scFv as a Potential Ligand for Carrying Drugs Targeting Selected Areas of the Brain"

_biomolecules, 2022, doi:10.3390/biom12060749_

Round 1

Reviewer 1 Report

Given that sufficient work has been done in original study and revision, the revised manuscript may be accepted for publication.

Reviewer 2 Report

The authors presented a study in scFv that specifically recognizes dopamine D2 and serotonin 5-HT1A receptor heteromers. The authors have revised the manuscript upon the reviewer’s comments. I have no further comments for the authors.

This manuscript is a resubmission of an earlier submission. The following is a list of the peer review reports and author responses from that submission.

Round 1

Reviewer 1 Report

The authors have performed all the necessary modifications and the manuscript can now be accepted for publication.

Author Response

Thank you for your comment.

Reviewer 2 Report

The presented version of the manuscript well described the potential ligand of Dopamine D2 and Serotonin 5-HT1A dimeric receptor-binding monomeric antibody scFv carries drugs targeting selected areas of the brain. The data content is considered a novelty and suitable for publication in Biomolecules. In addition to that, please see some of the minor suggestions.

1.

How will protein stability be affected by Ionic strength and pH?

2.

The conclusion is too short and has more scope to provide significant additions.

3.

References are not in the journal format; correct it.

Author Response

Ad1 In our experiments the strong influence of pH on protein (scFv) stability has occurred. On the other hand, the ionic strength didn’t show such strong effects.

Ad2 The conclusion has been modified.

Ad3 The references have been corrected.

Reviewer 3 Report

The manuscript attempts to investigate monomeric antibody scFv as a potential ligand to target Dopamine D2 and Serotonin 5-HT1A. A nanogold conjugate was also studied. The topic is important. However, the manuscript does not reach a publishable quality as it is not presented with a clear and well-organized manner. Much data presented are not significant in general and are lack of refining. There was no characterization of scFv-nanogold conjugate such as size, stability, conjugation rate, etc. There's no animal study. The study with BBB in in vitro model is questionable due to lack of proper controls (targeting ligand only, nanogold only, etc.). There's also no drug used for the study.

Author Response

Thank you for your comments. However, we cannot agree with them – in our opinion, the manuscript raises a very important issue. Of course, in vivo experiments are very desirable and will be done in the future. Obviously, we are perfectly aware, that the most important would be clinical studies, but one has to start from something, and experimental design in the present work deals with in vitro effects.

Reviewer 4 Report

The authors presented a study in scFv that specifically recognises dopamine D2 and serotonin 5-HT1A receptor heteromers. The scFv was then analyzed biochemically using CD spectroscopy. Functionally, the scFv can interact with the D2-5-HT1A receptor heteromer. The downstream cAMP signalling was also changed with the present of scFv. With gold-nanoparticle coating, the scFv can cross the blood-brain barrier in vitro. Overall, the experimental design and methods are clean and clear. I have few minor comments for the authors.  
  1. Can authors use the same method to calculate the ratio of structural motif (in table2) for 180 minute curve in figure3? That can further support the stability of scfv. Also is there any structure prediction from scFv sequence showing the beta sheet (table 2) is around 36-38%?
  2. Curves in Figure 3 is not as smooth as Figure 2A curve. Is there any reasons between them beyond the experimental conditions?
  3. Figure 7 caption seems not complete for the last sentence. “Errors are estimated using the standard devia-“. The caption should be clarified.
  4. Table 2 caption. There is a typo, "Structural motifs comparison”.

Author Response

Ad 1. Thank you for the suggestion. We calculated the structural motifs for the 180-minute curve (figure3) - appropriate data have been introduced in the manuscript. Similarly, data obtained from structure prediction - BLAST, Jpred, RaptorX tools - have been added to the main text.

Ad 2. No, there is not any other reasons.

Ad 3. It has been corrected

Ad 4. It has been corrected